# Water-Soluble Chalcogenide W_6_-Clusters: On the Way to Biomedical Applications

**DOI:** 10.3390/ijms23158734

**Published:** 2022-08-05

**Authors:** Alena D. Gassan, Anton A. Ivanov, Tatiana N. Pozmogova, Ilia V. Eltsov, Natalia V. Kuratieva, Yuri V. Mironov, Michael A. Shestopalov

**Affiliations:** 1Nikolaev Institute of Inorganic Chemistry of Siberian Branch of Russian Academy of Sciences, 3 Acad. Lavrentiev ave., 630090 Novosibirsk, Russia; 2Department of Natural Sciences, Novosibirsk State University, 1 Pirogova st., 630090 Novosibirsk, Russia

**Keywords:** octahedral chalcogenide tungsten clusters, crystal structure, water solubility, cytotoxicity

## Abstract

Despite the great potential of octahedral tungsten cluster complexes in fields of biomedical applications such as X-ray computed tomography or angiography, there is only one example of a water-soluble W_6_Q_8_-cluster that has been reported in the literature. Herein we present the synthesis and a detailed characterization including X-ray structural analysis, NMR, IR, UV–Vis spectroscopies, HR-MS spectrometry, and the electrochemical behavior of two new cluster complexes of the general formula W_6_Q_8_L_6_ with phosphine ligands containing a hydrophilic carboxylic group, which makes the complexes soluble in an aqueous medium. The hydrolytic stability of the clusters’ aqueous solutions allows us to investigate for the first time the influence of W_6_-clusters on cell viability. The results obtained clearly demonstrate their very low cytotoxicity, comparable to the least-toxic clusters presented in the literature.

## 1. Introduction

The discovery of X-rays by Wilhelm Conrad Röntgen and the printing of the first “medical” X-ray image of his wife’s hand in 1895 has resulted in a breakthrough in medical diagnostic procedures [1]. However, since X-rays are absorbed strongly by hard tissues (e.g., bones) and weakly by soft tissues (e.g., muscles, blood vessels), in 1896 radiocontrast media (bismuth, lead, and barium salts) were first utilized to precisely distinguish soft tissues. Nowadays, much safer radiopaque agents based on 1,3,5-triiodobenzene derivatives (e.g., iohexol, iopamidol, etc.) are used in medical practice [2]. However, the improvement of the parameters of contrast agents towards increasing their safety and image quality is still required [3,4,5,6]. An alternative way to solve this problem is to develop new highly hydrophilic and structurally stable compounds with high atomic numbers *Z*.

In review article by Yu and Watson [7] published in 1999, the authors mentioned that heavy metal clusters with different nuclearity are of great interest as components of X-ray contrast media. For example, di- and trinuclear tungsten clusters with cores {W_2_O_2_Q_2_} (Q = O or S), {W_3_Q_2_}, and {W_3_Q_4_}, grafted by polycarboxylic ligands such as ethylenediaminetetraacetate, triethylenetetramine-N,N,N′,N″,N‴,N‴-hexaacetate, ethyleneglycol-*bis*-(β-aminoethylether)-N,N,N′,N′-tetraacetate, 3,3′-[ethane-1,2-diylbis(oxy)]dipropanate, etc., have shown encouraging results in both in vitro and in vivo X-ray imaging studies providing high-contrast enhancement, with noticeable improvements in image quality along with suitably low acute toxicities in mice [7,8,9,10,11]. Nowadays, the chemistry of cluster compounds with such low nuclearity is being actively studied [12,13], but investigation from the point of view of application as radiocontrast agents is limited by the works noted above.

Metal cluster complexes with an octahedral cluster core, such as {M_6_X_8_} and {M_6_X_12_} (M = Ta, W, Re; X = halogen or chalcogen), represent another promising class of X-ray contrast agents. Due to the presence of six heavy metals and eight heavy inner ligands X (when X = I or Te), such clusters demonstrate multiplied X-ray attenuation relative to iodinated contrast agents [14,15,16,17,18,19]. At the moment, the octahedral cluster complexes most studied in terms of X-ray contrast agents are rhenium clusters with functionalized phosphines [{Re_6_Q_8_}(PR_3_)_6_]^n−^ (Q = S, Se; PR_3_ = P(CH_2_CH_2_CONH_2_)(CH_2_CH_2_COOH)_2_, P(CH_2_CH_2_COOH)_3_ or PPh_2_CH_2_CH_2_COOH). The compounds obtained are highly water-soluble, have low cellular and acute toxicities and can act as highly radiopaque agents for angiography and computed tomography [15,20,21,22]. In spite of all of the positive properties of rhenium clusters, the low world production of rhenium in cooperation with its high cost stimulates the study of the closest analogues of Re_6_Q_8_-clusters—octahedral tungsten cluster complexes with the general formula [{W_6_X_8_}L_6_], where X is halogen or chalcogen and L is terminal inorganic or organic ligands.

Halide, and especially iodide W_6_-clusters, have shown high X-ray attenuation, indicating perspectives for further study [14,16]. However, such clusters have two disadvantages. Firstly, they are hydrolytically unstable and form a precipitate of aquahydroxocomplexes [{W_6_X_8_}L_6-x-y_(H_2_O)_x_(OH)_y_] in aqueous media [16,23]. Secondly, {W_6_X_8_}-clusters can act as powerful photosensitizers of singlet-oxygen generation under X-ray irradiation [16,24]. Both of these disadvantages are a serious obstacle in the development of a radiopaque agent based on halide W_6_-clusters. It is important to note that some Re_6_-clusters can also act as moderate photosensitizers under X-ray irradiation, which also limits the development of contrast agents based on them [22,25].

Octahedral tungsten chalcogenide clusters [{W_6_Q_8_}L_6_] (Q—S, Se, Te) are free of both of these drawbacks. They are more hydrolytically stable than tungsten halides, and no examples of their photosensitizing properties are known to date. However, the pool of such clusters is not large enough due to the synthetic problems: only 36 individual cluster complexes (31 for Q = S, 2 for Q = Se, and 3 for Q = Te) with different organic or inorganic terminal ligands were described [26,27,28,29,30,31,32,33,34,35,36,37,38,39,40,41,42,43]. Moreover, only one of the known W_6_Q_8_-clusters is soluble in water, namely [{W_6_S_8_}(CN)_6_]^6−^ [38]. Hence, the preparation of new water-soluble W_6_Q_8_-based cluster complexes is still an important task.

In this work we report the synthesis detail characterization including single-crystal structure analysis; ^1^H, ^13^C, ^31^P, and ^77^Se NMR; the high-resolution mass spectrometric, electrochemical study of tungsten clusters with functionalized phosphine ligand PPh_2_CH_2_CH_2_COOH, namely neutral [{W_6_Q_8_}(PPh_2_CH_2_CH_2_COOH)_6_] (Q = S (H_6_-**1**), Se (H_6_-**2**)) and its water-soluble sodium salts Na_6_[{W_6_Q_8_}(PPh_2_CH_2_CH_2_COO)_6_] (Q = S (Na_6_-**1**) and Se (Na_6_-**2**)). Furthermore, stability in physiological medium and cytotoxic studies of water-soluble clusters were investigated.

## 2. Results and Discussion

### 2.1. Synthesis and Characterization

The compound H_6_-**1** was prepared according to the modified method published for the cluster with phosphine P(C_2_H_4_CN)_3_ as a ligand, namely [{W_6_S_8_}(P(C_2_H_4_CN)_3_)_6_] [43]. Briefly, the reaction mixture of (Bu_4_N)_2_[{W_6_I_8_}I_6_]/NaHS/^t^BuOK/PPh_2_CH_2_CH_2_COOH in a molar ratio of 1:16:8:16 in dry DMF was heated in a sealed glass tube at 100 °C for 4 days, resulting in the complete substitution of inner and terminal iodine ligands with a good yield (69%). Another synthetic approach was applied to obtain H_6_-**2** with a good yield (75%). Se^2−^-anions were generated in dry DMF under Ar atmosphere from elemental Se and NaBH_4_ in a molar ratio of 1:2, than a DMF solution of (Bu_4_N)_2_[{W_6_I_8_}I_6_] and PPh_2_CH_2_CH_2_COOH was added to the reaction mixture, keeping the molar ratio of the initial cluster, chalcogen and ligand, as well as reaction conditions, similar to sulfide analogue. After the reaction, the desired compounds were in the precipitate, which was washed successively with DMF and acetonitrile (to remove the residues of the reaction mixture and byproducts). The resulting powder was soluble in water (apparently due to the sodium salts formed during the synthesis). The solution was separated from the insoluble byproducts, and the final product was precipitated with a 0.1 M hydrochloric acid solution. In this case, only the complex precipitates were obtained, while all inorganic salts remained in solution and were removed by washing the complex with water. Therefore, this approach allowed us to obtain clusters of composition [{W_6_Q_8_}L_6_] in one step from the halide complex (Bu_4_N)_2_[{W_6_I_8_}I_6_] (without using [{W_6_S_8_}(TBP)_6_] or [{W_6_S_8_}(^n^BuNH_2_)_6_], which were applied in past works [33,34,35,39,40,41,42]). It should be noted that H_6_-**2** is the first example of a {W_6_Se_8_}-cluster with a phosphine ligand.

The composition of compounds was proven by elemental analysis and energy-dispersive X-ray spectroscopy (EDS). All characteristic vibrations of the ligand were observed in both compounds’ FTIR spectra (Appendix A). Moreover, the crystal structures of H_6_-**1** and H_6_-**2** as solvates with diethyl ether and water molecules (H_6_-**1**·7H_2_O·2.5Et_2_O and H_6_-**2**·3H_2_O·Et_2_O) were determined using single-crystal X-ray structural analysis (SCXRD).

Both compounds are isostructural to each other and to molybdenum analogs [{Mo_6_Q_8_}(PPh_2_CH_2_CH_2_COOH)_6_]·4.5H_2_O·2.5Et_2_O [44] and crystallize in triclinic space group *P*1 (Z = 1). The crystallographic data are summarized in Appendix A. The unit cell contains three independent W atoms and four S/Se atoms belonging to the same cluster unit. All atoms in the crystal structures are in general positions, while the center of the cluster coincides with the special crystallographic position (0, ½, 0) with C_i_ site symmetry. Phosphine ligands are coordinated to W atoms by a phosphorus atom (Figure 1). A similar type of coordination of PPh_2_CH_2_CH_2_COOH was previously reported for molybdenum and rhenium clusters, namely [{Mo_6_Q_8_}(PPh_2_CH_2_CH_2_COOH)_6_] and [{Re_6_Q_8_}(PPh_2_CH_2_CH_2_COOH)_6_]^2+^ [22,44]. An analysis of the bond lengths shows differences in the M–P distances for the cluster family [{M_6_Q_8_}(PPh_2_CH_2_CH_2_COOH)_6_]^n^ (M = Mo, W, Re) (Table 1). M–P bond length decreases, and hence bond strength increases in the order Mo—W—Re. Neutral clusters in structures H_6_-**1**·7H_2_O·2.5Et_2_O and H_6_-**2**·3H_2_O·Et_2_O are densely packed and participate in a number of strong hydrogen bonds between carboxylic groups and solvate molecules (O…O distances are 2.649–2.758) or between carboxylic acid groups of neighboring clusters (O…O distances are 2.642 and 2.634 for Q = S, Se correspondingly). The space between the clusters is occupied by various solvate molecules. No other remarkable interactions were observed in the crystal structures.

We also studied the compounds obtained in a DMSO-d_6_ solution using NMR spectroscopy. On ^1^H NMR spectra (Figure 2a), clusters’ phenyl protons are not shifted relative to the uncoordinated ligand, while methylene protons split into two groups with shifts (for P-C*H*_2_- Δ = 0.24 (overlap with DMSO signal) and 0.38 ppm, for -C*H*_2_-COO Δ = −0.15 and −0.32 ppm for H_6_-**1** and H_6_-**2**, correspondingly), which prove ligand coordination to the cluster core. Furthermore, both ^31^P NMR spectra (Figure 2b) contain one signal of coordinated PPh_2_CH_2_CH_2_COOH (shifted by 4.2 and −2.05 ppm for H_6_-**1** and H_6_-**2**, correspondingly), which confirms the symmetric coordination of apical ligands, while the presence of satellites with J = 230.7 and 208.6 Hz for H_6_-**1** and H_6_-**2**, correspondingly, prove W–P bonding. The shift of the phosphorus signal in opposite directions may indicate a different electronic configuration of the cluster core: the more electronegative sulfur withdraws the electron density from tungsten more strongly, due to which phosphorus donates more to the tungsten, while in the case of selenium the electron density on tungsten is higher and therefore, apparently, a slight donation from the metal to the phosphorus atom was observed. Moreover, for the first time for clusters with a {W_6_Se_8_} core, ^77^Se NMR spectroscopy was used for the characterization of H_6_-**2** (Figure 2c), indicating one signal at 1018 ppm.

### 2.2. Redox Properties

Octahedral chalcogenide transition metal clusters are known for their redox activity. Usually, {W_6_Q_8_}-clusters with phosphine ligands have CV curves with two reversible one-electron reduction and oxidation processes and two irreversible oxidation processes at high positive potentials. In [42] authors showed that [{W_6_S_8_}(PEt_3_)_6_], containing 20 metal-based valence electrons per cluster (VEC), possesses reversible reduction (E_1/2_ = −1.28 V vs. Ag/AgCl) from 20 to 21 VEC and oxidation (E_1/2_ = 0.1 V vs. Ag/AgCl) from 20 to 19 VEC. Molybdenum analogues [{Mo_6_Q_8_}(PEt_3_)_6_] (Q = S, Se) showed the same behavior but are shifted compared to {W_6_Q_8_}-clusters’ potentials—E_1/2_ = −1.05 V vs. Ag/AgCl for reduction and E_1/2_ = 0.22 V vs. Ag/AgCl for oxidation—with no difference between S and Se clusters [45].

In this work, the electrochemical properties of {W_6_S_8_} and {W_6_Se_8_} clusters with the same apical ligands were compared for the first time. Solutions of H_6_-**1** and H_6_-**2** in DMSO were studied by cyclic voltammetry (CV). According to CV curves (Figure 3 and Appendix A), both complexes demonstrate similar behavior and have reversible one-electron reduction with E_1/2_ = −0.9 V vs. Ag/AgCl, relating to transition from 20 to 21 VEC and reversible one-electron oxidation from 20 to 19 VEC at E_1/2_ = 0.39 and 0.35 V vs. Ag/AgCl for H_6_-**1** and H_6_-**2**, correspondingly. It is important to mportant to note that chalcogen in the cluster core does not significantly affect the redox potentials of compounds, indicating the metal-centered nature of the redox processes.

### 2.3. Water-Soluble Salts

Since neutral forms are only soluble in some organic solvents, those that are soluble in water anionic complexes were formed by the deprotonation of the carboxyl groups with alkali, resulting in hexasodium salts Na_6_-**1** and Na_6_-**2**. The composition of compounds was proven by elemental analysis and EDS, while FTIR for both salts contained all characteristic peaks of the pro-ligand with shifts for C=O vibration due to the deprotonation of the carboxyl group (Appendix A).

For Na_6_-**1**, the single crystal suitable for X-ray structural analysis was obtained by acetone diffusion in an aqueous/ethanol solution of the cluster. According SCXRD, Na_6_-**1**·7.5H_2_O·Me_2_CO crystalizes in triclinic space group *P*1 (Z = 1). The crystallographic data are summarized in Appendix A. The unit cell contains three independent W atoms and four S atoms belonging to the same cluster. All atoms in the crystal structure are in general positions, while the center of the cluster coincides with the centrosymmetric special crystallographic position (½, ½, 0). As well as for neutral cluster H_6_-**1**, in Na_6_-**1**, the coordination of the phosphine ligand to the cluster core by the P atom is preserved. The W–P bond lengths are slightly longer than those for H_6_-**1** (see Table 1).

The compound Na_6_-**1**·7.5H_2_O·Me_2_CO contains six Na^+^, which compensates for the high negative charge of the cluster anion [{W_6_S_8_}(PPh_2_CH_2_CH_2_COO)_6_]^6−^. Each oxygen atom of the carboxylic group of the ligand (except O22) interacts with alkali cations, partially filling their coordination sphere, while molecules of water and acetone occupy the rest of the coordination spheres. The remaining disordered water molecule (O6W) unassociated with sodium cations is involved in hydrogen bonds with the carboxylic groups of the ligands (O…O distances of 2.717 and 3.094 Å). The solvated alkali cations are island-like packed, and link two cluster anions with the formation of infinite chains—{alkali cations-cluster anions}^∞^ (Appendix A). These chains are stacked in parallel and do not interact with neighboring ones. However, there is still a volume available for the solvent in the crystal structure between the chains. According to the PLATON software [46,47], a void with the volume of 553 Å^3^ and about 111 electrons was found (Appendix A). This electron density could be attributed to 11 water molecules occupying a minimum 440 Å^3^ or other solvent molecules, i.e., acetone or ethanol. No other remarkable interactions were observed in the crystal structure.

^1^H and ^31^P NMR spectroscopy were used to confirm the preservation of clusters in alkali solutions (Appendix A). Thus, there are shifts for both phenyl (Δ = 0.13–0.18 and 0.15–0.18 ppm for Na_6_-**1** and Na_6_-**2**, correspondingly) and methylene (for P-C*H*_2_- Δ = 0.22 and 0.35 ppm and for -C*H*_2_-COO Δ = −0.26 and −0.37 ppm for Na_6_-**1** and Na_6_-**2**, correspondingly) protons in the ^1^H NMR spectra that indicates the coordination of PPh_2_CH_2_CH_2_COOH. The ^31^P NMR spectra contain only one shifted signal for each compound (Δ = 5.31 and −2.76 ppm), with satellites of W–P bonding with coupling constants 231.4 and 208.2 Hz for Na_6_-**1** and Na_6_-**2**, correspondingly. As well as H_6_-**2**, Na_6_-**2** has only one signal at 998 ppm in the ^77^Se NMR spectrum in a wide field range (Appendix A).

Furthermore, according to high-resolution mass spectrometry, water solutions of Na_6_-**1** and Na_6_-**2** contain cluster forms with six ligands and various cationic compositions (Figure 4 and Appendix A), additionally indicating the existence of clusters in aqueous solutions.

### 2.4. Stability in Culture Medium

To the best of our knowledge, only (Alk)_6_[{W_6_S_8_}(CN)_6_] (Alk = Na, K) is water-soluble among all {W_6_Q_8_}-clusters. Jin et al. [38] showed the stability of this cluster in deoxygenated water using ^13^C NMR spectroscopy. In this work, we studied the behavior of water-soluble Na_6_-**1** and Na_6_-**2** in culture medium (Dulbecco’s Modified Eagle’s Medium, DMEM) by UV–Vis spectroscopy. As can be seen in Figure 5, spectra profiles were unchanged, while absorptions remained almost at the same level after 72 h for both salts, which, in total, proved the hydrolytic stability of the clusters.

### 2.5. Cytotoxicity

The MTT assay is a classic test for estimating cytotoxicity. It is based on the reduction of a bright yellow MTT reagent to violet formazan by the mitochondrial activity of the cells. However, in this work, we demonstrate the ability of the clusters to reduce MTT by themselves. After the mixing of the Na_6_-**1** or Na_6_-**2** solutions with the solution of the MTT reagent, a violet, poorly soluble in water but highly soluble in ^i^PrOH precipitate is rapidly formed. A broad signal in the range of 500–600 nm appeared on the UV–Vis spectra of aqueous solutions after reactions (Appendix A), in which the profile and position coincide with the UV–Vis spectrum of the violet precipitate in ^i^PrOH and with the literature data for formazan. At the same time, no changes were observed in the profiles and positions of the clusters’ signals. Such an observation can be explained by the redox properties of clusters, which, as demonstrated above, almost does not depend on chalcogen in the cluster core. Since the cluster complex was in excess with respect to the MTT reagent (molar ratio ∽4:1), no significant changes in the absorption spectra were found. Despite the fact that the reduction of the MTT reagent by various compounds is not new [48,49], such a result has not been previously reported for octahedral cluster complexes of transition metals.

Due to the data obtained, the cytotoxicity studies were carried out using double staining with Hoechst 33342/propidium iodide (PI) (Figure 6). Due to the fact that propidium iodide penetrates only into dead cells with impaired membrane integrity, and Hoechst 33342 stains the chromatin of all cells, the double-staining method allows us to calculate the percentage of living and dead cells for each investigated concentration of the Na_6_-**1** or Na_6_-**2**. In addition, this method also allows one to distinguish apoptotic cells. In apoptotic cells, chromatin is condensed, and, while staining with Hoechst 33342, a brighter signal is observed. Na_6_-**1** and Na_6_-**2** were incubated with Hep-2 cells for 48 h. The IC_50_ values for the clusters synthesized here in comparison with those for rhenium analogues are presented in Table 2. It should be noted that such studies for tungsten chalcogenide cluster compounds were carried out for the first time.

The difference in the toxicity of tungsten and rhenium clusters can be associated with different charges of complexes, which leads to different charge distribution and osmolality and hence, different chaotropic effect (specific ion effect) [50,51] of the clusters. However, for both types of the clusters, an increase in toxicity was observed while changing the inner ligand from sulfur to selenium. Indeed, a series of cluster complexes, [{Re_6_Q_8_}L_6_] (varying L), was previously shown to possess different biological effects: cytotoxicity, cellular uptake, and acute toxicity in mice depending on inner ligand [18,20,21,22,52,53,54,55]. For example, this tendency was observed in the cytotoxic effects of clusters with different ligand natures, namely [{Re_6_Q_8_}(CN)_6_]^4−^ (Q = S, Se, Te), [17,18] [{Re_6_Q_8_}(trz)_6_]^4−^, [55] [{Re_6_Q_8_}(PPh_2_CH_2_CH_2_COO)_6_]^4−^ [22] and [{Re_6_Q_8_}(P(CH_2_CH_2_COO)_3_)_6_]^16−^ [21] (Q = S, Se; trz = 1,2,3-triazole, 1,2,4-triazole or benzotriazole), studied in normal and cancerous cell lines. Specifically, in the case of clusters with acidic ligands, like triazoles and phosphines, the selenium-containing clusters have a more pronounced cytotoxic effect than sulfur ones. This behavior is obviously related to the stronger electron-accepting properties of sulfur atoms in the cluster core than those of selenium ones, which lead to the higher acidity of terminal ligands. Moreover, the higher acidity of the ligands increases the charge of the cluster in a solution, resulting in a lower chaotropic effect (due to the higher charge density) of the complex and, at the same time, a lower cytotoxicity. On the other hand, chaotropic agents are known to disrupt the hydrogen bonding network and to reduce the stability of the native state of intracellular proteins by weakening the hydrophobic effect [56]. Toxicology research and investigations of chaotropic effects for [{Re_6_Q_8_}(CN)_6_]^4−^ (Q = S, Se, Te) are one of the examples of links between specific ion effects and cytotoxicity. The least-toxic cluster complex [{Re_6_Te_8_}(CN)_6_]^4−^ is the least chaotropic agent, while the more toxic [{Re_6_S_8_}(CN)_6_]^4−^ is the more chaotropic [17,18,57,58,59]. Apparently, in the case of the highest chaotropic effect, the strong supramolecular binding of clusters to various intracellular organelles and biomolecules is possible, while the least chaotropic compounds practically do not participate in any interactions. In this work, according to ^31^P NMR data (Appendix A), different electron densities on phosphorus atoms (different chemical shifts) confirm the effect of chalcogen in the cluster core on the acidic properties of the carboxylic groups of ligands and, therefore, the chaotropic effect of the clusters, which is probably the reason for the higher cytotoxicity of Na_6_-**2**.

While lower toxicity was shown for Na_6_-**1**, Na_6_-**2** can also be considered a promising compound due to its tendency to cause cell death through the mechanism of apoptosis. Unlike necrosis, ferroptosis, and autophagy, apoptosis does not cause a significant inflammatory response in the body [60]. In concentrations from 19.5 to 156.3 μM, one can see the predominance of apoptotic cells over cells with impaired membrane integrity (stained with propidium iodide). This suggests that the Na_6_-**2** in this concentration range causes a sufficient toxic effect to cause apoptosis but not enough to lead to membrane destruction and necrotic cell death. At the moment, it is impossible to explain exactly the reason for this effect, since apoptosis can be caused by both extracellular and intracellular (mitochondrial) signaling pathways [60]. More detailed study of this process is a goal of our further research.

To confirm that Na_6_-**2** is able to initiate the apoptosis of tumor cells, an additional study was conducted. Hep-2 cells were incubated with Na_6_-**2** at concentrations of 78.1, 156.3, and 312.5 μM and stained with fluorescein isothiocyanate (FITC) conjugated with Annexin V and propidium iodide (PI). PI enters the cell if the cellular membrane is disrupted, binds to the DNA, and becomes intensely fluorescent, indicating necrosis. Annexin V binds to phosphatidylserine, which is a marker of apoptosis on the plasma membrane’s outer layer. Thus, cells stained only with Annexin V indicate an early stage of apoptosis, while a double-positive signal shows late apoptosis.

In the population of apoptotic cells, late apoptotic cells predominate in all three concentrations studied. With such a long incubation time (24 h), it is expected that cells have already passed the stage of early apoptosis and now enter late apoptosis. One can see in Figure 7 that the total number of early and late apoptotic cells at a concentration of 78.1 μM is 20.7% (green A+ PI− and purple A+ PI+ squares). At a concentration of 156.3 μM, the number of apoptotic cells increases to 26.9%, and the number of necrotic cells also increases to 11.1%. At a concentration of 312.5 μM, the population of necrotic cells prevails over the population of apoptotic cells, reaching 23.9% (red A− PI+ square). It is important to note that the concentration of dead cells was even higher, because these cells could be lost during the washing of the samples. Despite the predominance of necrotic cells, the number of apoptotic cells in the concentration of 312.5 μM is still high and reaches 18.4%. These results correlate with the data obtained in the double-staining experiment. Thus, we can state that Na_6_-**2** causes the concentration-dependent initiation of apoptosis in cancer cells. The peak concentration for this effect is at 156.3 μM. A lower concentration causes a less pronounced cellular response, while a higher concentration has a stronger toxic effect, causing necrosis.

## 3. Materials and Methods

### 3.1. Materials

All reagents and solvents employed were commercially available and used as received without further purification. (Bu_4_N)_2_[{W_6_I_8_}I_6_] was prepared according to the procedure described in [16].

### 3.2. Methods

Elemental analyses were obtained using a EuroVector EA3000 Elemental Analyser (EuroVector S.p.A., Milan, Italy). FTIR spectra were recorded on a Bruker Vertex 80 (Bruker Optics GmbH & Co. KG, Ettlingen, Germany) as KBr disks. Energy-dispersive X-ray spectroscopy (EDS) was performed on a Hitachi TM3000 TableTop SEM (Hitachi High-Technologies Corporation, Tokyo, Japan) with Bruker QUANTAX 70 EDS equipment. Thermal properties were studied on a Thermo Microbalance TG 209 F1 Iris (NETZSCH) from 25 to 800 °C, at a rate of 10 °C·min^−1^ in He flow (30 mL/min). The absorption spectra were investigated using an Agilent Cary 60 UV–Vis spectrophotometer (USA). High-resolution electrospray mass spectrometric (HR-ESI-MS) detection was performed at the Center of Collective Use “Mass spectrometric investigations” SB RAS in negative mode within the 500–3000 *m*/*z* range on an electrospray ionization quadrupole time-offlight (ESI-q-TOF) high-resolution mass spectrometer Maxis 4G (Bruker Daltonics, Bremen, Germany).

#### 3.2.1. NMR Studies

The 1D and 2D NMR spectra of salts and acids were recorded from a D_2_O and DMSO solutions, respectively, at room temperature on a Bruker Avance III 500 FT-spectrometer (Bruker BioSpin AG, Faellanden, Switzerland) with working frequencies 500.03, 125.73, 202.42, and 95.36 MHz for ^1^H, ^13^C, ^31^P, and ^77^Se nuclei, respectively. The ^1^H and ^13^C NMR chemical shifts are reported in ppm of the δ scale and referred to DSS signal as an internal standard for water solutions (δ(^1^H, ^13^C) = 0.0 ppm) and to the signals of the methyl group for DMSO solutions (δ(^1^H) = 2.50 ppm, δ(^13^C) = 39.50 ppm). Chemical shifts for the ^31^P spectra are referred to the external standard of 85% H_3_PO_4_ (δ(^31^P) = 0.0 ppm). The ^77^Se chemical shifts are referred to the external standards of a 1M SeO_2_ solution in D_2_O (δ(^77^Se) = 1282 ppm). The assignment of the signals was carried out using 2D ^1^H, ^13^C-HMBC NMR techniques.

#### 3.2.2. CV Studies

Cyclic voltammetry was carried out with an Elins P-20X8 voltammetry analyzer (Electrochemical Instruments, Chernogolovka, Russia) using a three-electrode scheme with GC working, Pt auxiliary, and Ag/AgCl/3.5M KCl reference electrodes. Investigations were carried out for 5·10^−4^M solutions of corresponding cluster compounds in 0.1 M Bu_4_NClO_4_ in DMSO under Ar atmosphere.

#### 3.2.3. Crystallography

Single-crystal X-ray diffraction data were collected using a Bruker Nonius X8 Apex 4K CCD diffractometer with graphite monochromatized MoKα radiation (λ = 0.71073 Å). Absorption corrections were made empirically using the SADABS program [61]. The structures were solved by the direct method and further refined by the full-matrix least-squares method using the SHELXTL program package [61,62]. All non-hydrogen atoms were refined anisotropically. The hydrogen atoms of solvate water molecules, as well as the acidic hydrogen atoms of (2-carboxyethyl)diphenylphosphane ligands in H_6_-**1**·7H_2_O·2.5Et_2_O, were not located. Appendix A summarizes crystallographic data, while CCDC 2190918-2190920 contains the supplementary crystallographic data for this paper.

#### 3.2.4. Cell Culture

The human larynx carcinoma cell line (Hep-2) was purchased from the State Research Center of Virology and Biotechnology VECTOR and cultured in Eagle’s Minimum Essential Medium (EMEM, pH = 7.4) supplemented with a 10% fetal bovine serum under a humidified atmosphere (5% CO_2_ and 95% air) at 37 °C.

#### 3.2.5. Viability, Apoptosis and Proliferation Assay

Cell viability, apoptosis, and proliferation were detected by Hoechst 33342/PI staining as previously described by Lee et al. [63] The Hep-2 cells were seeded on 96-well plates at 5 × 10^3^ cells per well in a medium containing Na_6_[{W_6_Q_8_}(Ph_2_PC_2_H_4_COO)_6_] at concentrations from 4.9 μM to 2.5 mM and incubated for 48 h. The cells incubated in the absence of cluster compounds were used as a control. Treated cells and control cells were stained with Hoechst 33342 (Sigma-Aldrich, Saint Louis, MO, USA) for 15 min at 37 °C and PI (Sigma-Aldrich, Saint Louis, MO, USA) for 10 min at 37 °C. An IN Cell Analyzer 2200 (GE Healthcare, Chalfont Saint Giles, UK) was used to perform the automatic imaging of six fields per well under 200× magnification, in brightfield and fluorescence channels. The images produced were used to analyze live, apoptotic, and dead cells among the whole population using the IN Cell Investigator software (GE Healthcare, UK).

#### 3.2.6. Flow Cytometric Analysis of Cell Death

A flow cytometry analysis was applied to quantify the ratio of live, early-apoptotic, late-apoptotic, and dead-cell populations. Hep-2 cells were seeded into 48-well plates at a starting density of 8 × 10^3^/well. Cells were treated with 78.1, 156.3, or 312.5 µM of Na_6_-**2** for 24 h. A FITC-Annexin V Apoptosis Detection Kit with PI (Bio-Rad, Hercules, CA, USA) was used to label cells according to the manufacturer’s instructions. The samples were measured with a CytoFLEX Flow Cytometer (Beckman, Brea, CA, USA). Double-negative (Annexin V−/PI−) cells were considered live. Annexin V positive (Annexin V+/PI−) and double-positive (Annexin V+/PI+) cells were identified as early and late apoptotic, respectively. PI positive cells (Annexin V−/PI+) were identified as necrotic. In order to preserve the dead cells when washing the samples, the medium with dead cells was collected from a 48-well plate and centrifuged separately at 1000 rpm, after which the resulting precipitate was added to the suspension of living cells at the time of staining the samples.

### 3.3. Synthetic Procedures

#### 3.3.1. Synthesis of [{W_6_S_8_}(Ph_2_PC_2_H_4_COOH)_6_] (H_6_-1)

A total of 150 mg (0.044 mmol) of (Bu_4_N)_2_[{W_6_I_8_}I_6_], 40 mg (0.713 mmol) of NaHS, 40 mg (0.357 mmol) of ^t^BuOK, and 184 mg (0.713 mmol) of Ph_2_PC_2_H_4_COOH were dissolved in 3 mL of dry DMF. The reaction mixture was heated at 100 °C for 96 h in a sealed glass tube and cooled to room temperature. The resulting precipitate was washed with acetonitrile until the filtrate was colorless, then it was dissolved in water, precipitated with 0.1 M HCl, and dried in air after washing twice with water. Yield: 89 mg (69%). Anal. Calcd. for [{W_6_S_8_}(Ph_2_PC_2_H_4_COOH)_6_]: C, 37.2; H, 3.1; S, 8.8; found C, 37.5; H, 3.2; S, 8.6. EDS: W/S/P atomic ratio was equal to 6:8.2:6.3. FTIR (KBr, cm^−1^): ν(C=O)—1705 and all the other expected peaks for the phosphine ligand without any significant shifts were observed. ^1^H NMR (500 MHz, DMSO): δ 2.06–2.16 (m, 2H, -C**H**_2_-COO), 2.49 (m, 2H, -C**H**_2_-P), 7.26–7.36 (m, 6H, *o*,*p*-Ph), 7.53 (t, 4H, J = 7.8 Hz, *m*-Ph). ^13^C{^1^H} NMR (126 MHz, DMSO): δ 26.83 (d, ^1^J_CH_ = 129 Hz, ^1^J_PC_ = 25.1 Hz, -**C**H_2_-P), 28.78 (s, ^1^J_CH_ = 129 Hz, -**C**H_2_-COO), 127.56 (d, ^1^J_CH_ = 160 Hz, ^3^J_PC_ = 8.8 Hz, *o*-Ph), 128.99 (s, ^1^J_CH_ = 160 Hz, *p*-Ph), 132.58 (d, ^1^J_CH_ = 161 Hz, ^2^J_PC_ = 10.6 Hz, *m*-Ph), 136.83 (d, ^1^J_PC_ = 35.6 Hz, *i*-Ph), 173.48 (d, ^3^J_PC_ = 16.0 Hz, **C**OO) (Appendix A). ^31^P NMR (202 MHz, H_3_PO_4_): δ −12.25 (J_PW_ = 230.7 Hz). The TG analysis indicated stability up to 230 °C (Appendix A). UV–Vis: λ = 412 nm, ε = 12567 M^−1^·cm^−1^ (Appendix A). Single crystals of H_6_-**1**·7H_2_O·2.5Et_2_O suitable for X-ray structural analyses were obtained with the slow diffusion of diethyl ether in the ethanol solution of H_6_-**1**.

#### 3.3.2. Synthesis of [{W_6_Se_8_}(Ph_2_PC_2_H_4_COOH)_6_] (H_6_-2)

A total of 75 mg (0.951 mmol) of Se was placed in a glass tube containing 2 mL of dry DMF. The mixture was heated to 70 °C under Ar atmosphere, and 72 mg (1.902 mmol) of NaBH_4_ was added. After the resulting dark red solution was discolored, 1 mL of dry DMF containing 200 mg (0.059 mmol) of (Bu_4_N)_2_[{W_6_I_8_}I_6_] and 245 mg (0.951 mmol) of Ph_2_PC_2_H_4_COOH was added. The reaction mixture in a sealed glass tube was heated at 100 °C for 96 h. The resulting precipitate was washed with DMF and acetonitrile until the filtrates were colorless, then it was dissolved in water, precipitated with 0.1 M HCl, and dried in air after washing thrice with ethanol. Yield: 145 mg (75%). Anal. Calcd. for [{W_6_Se_8_}(Ph_2_PC_2_H_4_COOH)_6_]: C, 32.9; H, 2.8; found C, 32.6; H, 2.9. EDS: W/Se/P atomic ratio was equal to 6:8.9:6.6. FTIR (KBr, cm^−1^): ν(C=O)–1708 and all the other expected peaks for the phosphine ligand without any significant shifts were observed. ^1^H NMR (500 MHz, DMSO): δ 1.96 (dt, 2H, ^4^J_PH_ = 8.0 Hz, ^3^J_HH_ = 6.5 Hz, -C**H**_2_-COO), 2.60–2.70 (m, 2H, -C**H**_2_-P), 7.29–7.37 (m, 6H, *o*,*p*-Ph), 7.37–7.43 (m, 4H, J = 7.8 Hz, *m*-Ph). ^13^C{^1^H} NMR (126 MHz, DMSO): δ 29.50 (s, ^1^J_CH_ = 129.8 Hz, -**C**H_2_-COO), 30.09 (d, ^1^J_CH_ = 129.8 Hz, ^1^J_PC_ = 24.8 Hz, -**C**H_2_-P), 127.50 (d, ^1^J_CH_ = 161.6 Hz, ^3^J_PC_ = 8.6 Hz, *o*-Ph), 128.94 (s, ^1^J_CH_ = 160.0 Hz, *p*-Ph), 132.40 (d, ^1^J_CH_ = 160.0 Hz, ^2^J_PC_ = 9.6 Hz, *m*-Ph), 138.61 (d, ^1^J_PC_ = 35.8 Hz, *i*-Ph), 173.41 (d, ^3^J_PC_ = 15.0 Hz, **C**OO) (Appendix A). ^31^P NMR (202 MHz, H_3_PO_4_): δ −18.53 (J_PW_ = 208.6 Hz). ^77^Se (95 MHz, SeO_2_) δ 1018. The TG analysis indicated stability up to 230 °C (Appendix A). UV–Vis: λ = 486 nm, ε = 9316 M^−1^·cm^−1^ (Appendix A). Single crystals of H_6_-**2**·3H_2_O·Et_2_O suitable for X-ray structural analyses were obtained by the slow diffusion of diethyl ether in the acetone solution of H_6_-**2**.

#### 3.3.3. General Synthetic Procedure for Na_6_[{W_6_Q_8_}(Ph_2_PC_2_H_4_COO)_6_] (Na_6_-1, Na_6_-2)

A total of 128 mg of H_6_-**1** or 145 mg of H_6_-**2** (0.044 mmol) were dissolved in 5 mL of water containing 12 mg of NaOH (molar equivalents are 1:7). The target products were precipitated with acetone, washed twice with ethanol to remove excess alkali, and dried in air.

Na_6_-**1**. Yield: 128 mg (95%). Anal. Calcd. for Na_6_[{W_6_S_8_}(Ph_2_PC_2_H_4_COO)_6_]: C, 35.5; H, 2.8; S, 8.4; found C, 35.3; H, 2.7; S, 8.2. EDS: W/S/P/Na atomic ratio was equal to 6:8.1:6.3:6.3. FTIR (KBr, cm^−1^): ν(C=O)–1573 and all the other expected peaks for the phosphine ligand without any significant shifts were observed. ^1^H NMR (500 MHz, DSS): δ 2.00 (m, 2H, -C**H**_2_-COO), 2.55 (m, 2H, -C**H**_2_-P), 7.26-7.37 (m, 6H, *o*,*p*-Ph), 7.53 (t, 4H, J = 7.5 Hz, *m*-Ph). ^13^C NMR (126 MHz, DSS): δ 31.38 (d, ^1^J_PC_ = 24.6 Hz, -**C**H_2_-P), 34.87 (s, -**C**H_2_-COO), 130.57 (d, ^3^J_PC_ = 8.8 Hz, *o*-Ph), 132.15 (s, *p*-Ph), 135.65 (d, ^2^J_PC_ = 10.3 Hz, *m*-Ph), 139.52 (d, ^1^J_PC_ = 36.7 Hz, *i*-Ph), 183.89 (d, ^3^J_PC_ = 14.6 Hz, **C**OO) (Appendix A). ^31^P NMR (202 MHz, H_3_PO_4_): δ −10.69 (J_PW_ = 231.4 Hz). The TG analysis indicated stability up to 310 °C (Appendix A). HR-ESI-MS (–): 975.6430 ([NaH_2_{W_6_S_8_}(PPh_2_C_2_H_4_COO)]^3−^), 982.9703 ([Na_2_H{W_6_S_8_}(PPh_2_C_2_H_4_COO)]^3−^), 990.2976 ([Na_3_{W_6_S_8_}(PPh_2_C_2_H_4_COO)]^3−^), 1485.9505 ([Na_3_H{W_6_S_8_}(PPh_2_C_2_H_4_COO)]^2−^), 1496.9414 ([Na_4_{W_6_S_8_}(PPh_2_C_2_H_4_COO)]^2−^). Single crystals of Na_6_-**1**·7.5H_2_O·Me_2_CO suitable for X-ray structural analyses were obtained by the slow diffusion of acetone in the aqueous/ethanol solution of Na_6_-**1**.

Na_6_-**2**. Yield: 142 mg (94%). Anal. Calcd. for Na_6_[{W_6_Se_8_}(Ph_2_PC_2_H_4_COO)_6_]: C, 31.6; H, 2.5; found C, 30.6; H, 2.7. EDS: W/Se/P/Na atomic ratio was equal to 6:8.3:6.1:6.2. FTIR (KBr, cm^−1^): ν(C=O)–1570 and all the other expected peaks for the phosphine ligand without any significant shifts were observed. ^1^H NMR (500 MHz, DSS): δ 1.83–1.95 (m, 2H, -C**H**_2_-COO), 2.60–2.77 (m, 2H, -C**H**_2_-P), 7.26–7.42 (m, 6H, *o*,*p*-Ph), 7.53 (t, 4H, J = 8.0 Hz, *m*-Ph). ^13^C{^1^H} NMR (126 MHz, DSS): δ 34.53 (d, ^1^J_PC_ = 24.6 Hz, -**C**H_2_-P), 35.60 (s, -**C**H_2_-COO), 130.34 (d, ^3^J_PC_ = 8.6 Hz, *o*-Ph), 131.95 (s, *p*-Ph), 135.52 (d, ^2^J_PC_ = 9.61 Hz, *m*-Ph), 141.83 (d, ^1^J_PC_ = 36.2 Hz, *i*-Ph), 183.86 (d, ^3^J_PC_ = 13.7 Hz, **C**OO) (Appendix A). ^31^P NMR (202 MHz, H_3_PO_4_): δ −18.76 (J_PW_ = 208.2 Hz). ^77^Se NMR (95 MHz, SeO_2_): δ 998. The TG analysis indicated stability up to 220°C (Appendix A). HR-ESI-MS (–): 1093.8371 ([H_3_{W_6_Se_8_}(PPh_2_C_2_H_4_COO)]^3−^), 1101.1644 ([NaH_2_{W_6_Se_8_}(PPh_2_C_2_H_4_COO)]^3−^), 1108.4917 ([Na_2_H{W_6_Se_8_}(PPh_2_C_2_H_4_COO)]^3−^), 1641.2595 ([H_4_{W_6_Se_8_}(PPh_2_C_2_H_4_COO)]^2−^), 1652.2505 ([NaH_3_{W_6_Se_8_}(PPh_2_C_2_H_4_COO)]^2−^), 1663.2415 ([Na_2_H_2_{W_6_Se_8_}(PPh_2_C_2_H_4_COO)]^2−^).

## 4. Conclusions

To conclude, in this work, two new chalcogenide octahedral tungsten clusters with phosphine ligands were obtained and characterized. Experimental data have shown that the compounds are redox-active, showing reversible oxidation and reduction without the destruction of the structure. Complexes can be readily dissolved in an alkaline aqueous solution (due to the presence of six carboxyl groups) and demonstrate high stability towards hydrolysis. Besides the known (Na/K)_6_[{W_6_S_8_}(CN)_6_] cluster, the compounds obtained are the only examples of water-soluble complexes of such a type. The high stability of the clusters in cultural medium allowed us to investigate, for the first time for octahedral tungsten clusters, the cytotoxicity on cancerous cell line Hep-2. The inner-ligand-dependent cytotoxicity observed is presumably due to specific ion (chaotropic) effects arising from the different acidity properties of the terminal ligands. To date, the cluster complex Na_6_[{W_6_S_8_}(PPh_2_CH_2_CH_2_COO)_6_] is one of the least-cytotoxic among all studied octahedral clusters, which opens up horizons for further investigation on small laboratory animals in terms of radiopaque media. For Na_6_-**2**, the ability for the concentration-dependent initiation of apoptosis in Hep-2 cancer cells was shown. This feature has been demonstrated for the first time for cluster complexes and might be a perspective for using this substance as a theranostic agent.

## Figures and Tables

**Figure 1 ijms-23-08734-f001:**
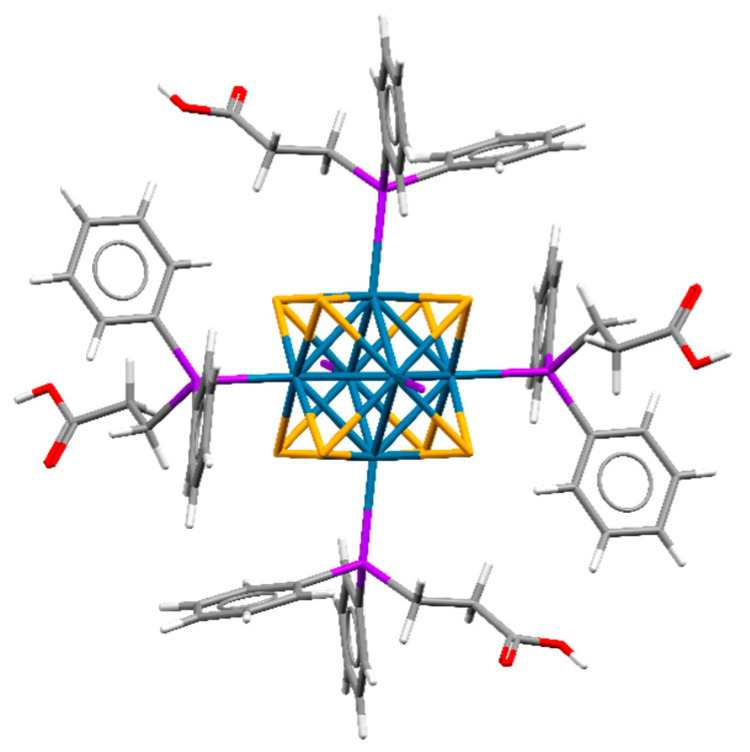
Structure of neutral cluster complexes [{W_6_Q_8_}(PPh_2_C_2_H_4_COOH)_6_]. Two phosphine ligands are omitted for clarity. W—blue, Q—orange, P—purple, C—grey, H—white, O—red.

**Figure 2 ijms-23-08734-f002:**
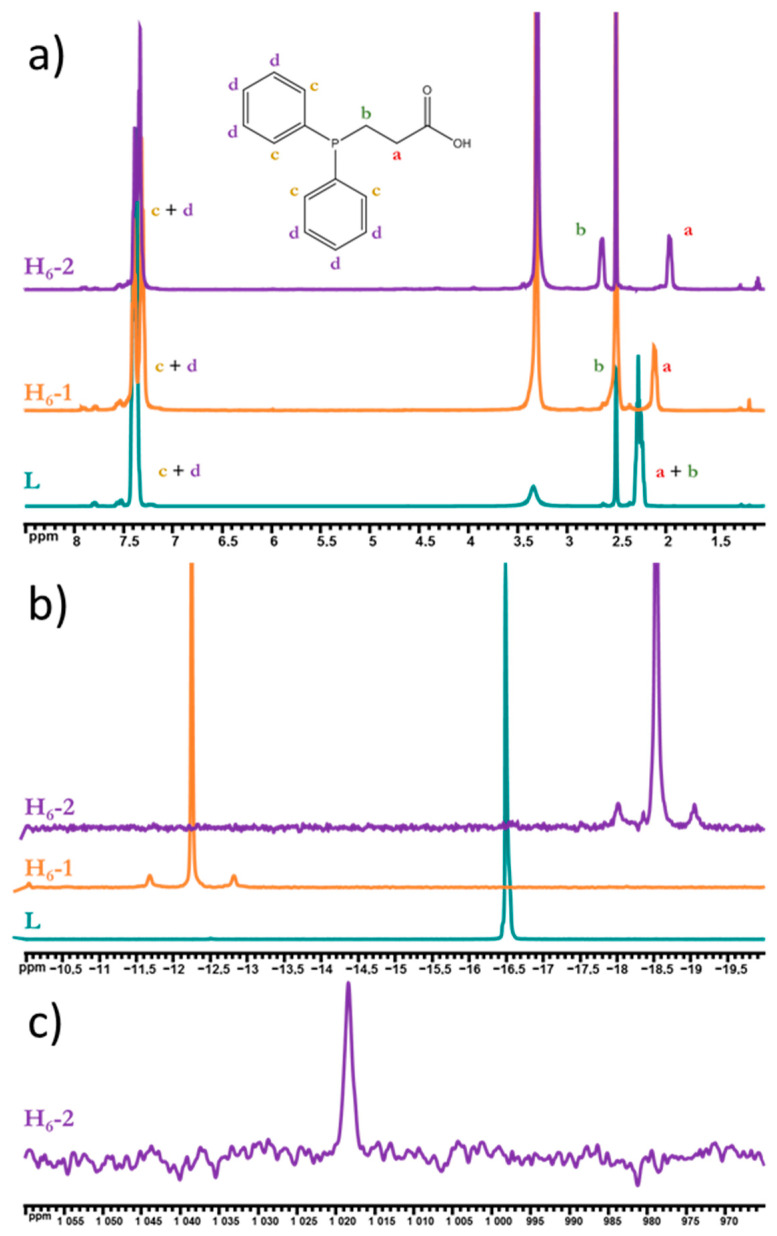
^1^H (**a**), ^31^P (**b**), ^77^Se (**c**), and NMR spectra of H_6_-**1** and H_6_-**2** in DMSO-d_6_ in comparison to PPh_2_C_2_H_4_COOH.

**Figure 3 ijms-23-08734-f003:**
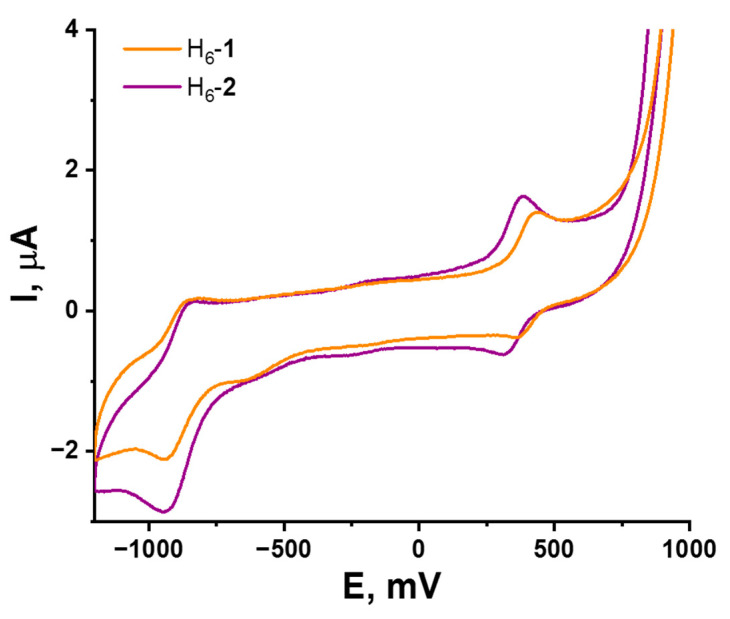
Cyclic voltammetry of H_6_-**1** and H_6_-**2** (0.5 mM) in 0.1 M Bu_4_NClO_4_ DMSO solution; scan rate—100 mV/s. Reference electrode: Ag/AgCl/3.5 M KCl.

**Figure 4 ijms-23-08734-f004:**
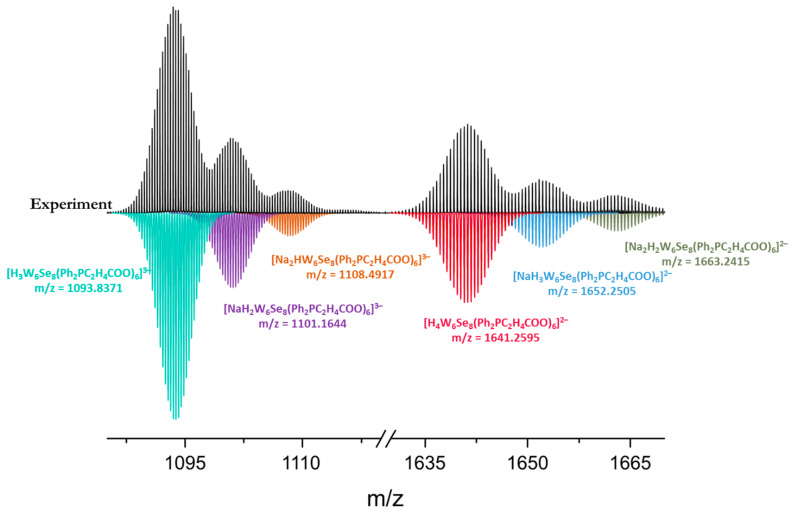
Fragments of the mass spectrum of Na_6_-**2** in aqueous solution (black) and simulated profiles of forms (colored).

**Figure 5 ijms-23-08734-f005:**
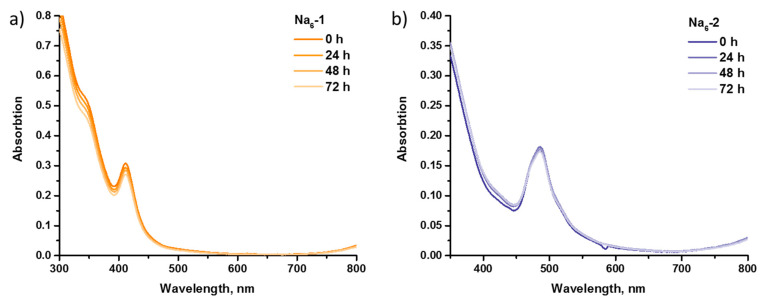
UV–Vis spectra of Na_6_-**1** (**a**) and Na_6_-**2** (**b**) in time in culture medium.

**Figure 6 ijms-23-08734-f006:**
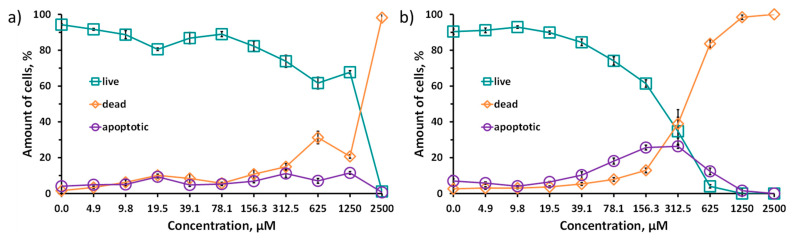
Viability of Hep-2 cells treated with Na_6_-**1** (**a**) and for Na_6_-**2** (**b**), determined by double-staining with Hoechst 33342/PI.

**Figure 7 ijms-23-08734-f007:**
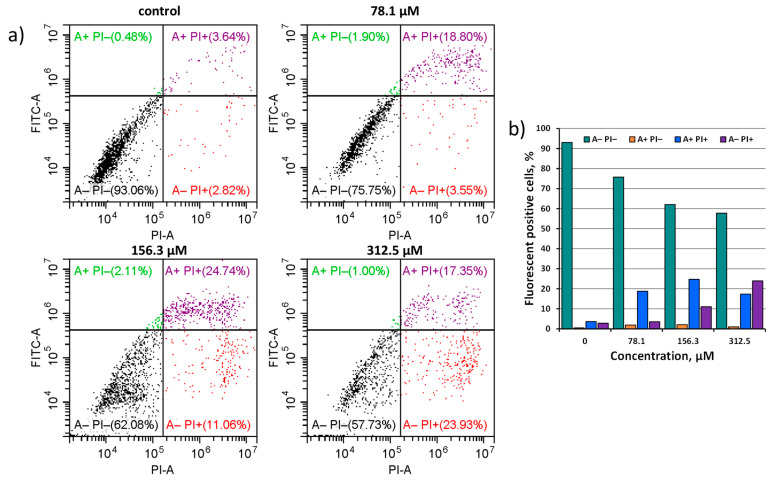
(**a**) Effect of Na_6_-**2** on the apoptosis of Hep-2 cells. Cells were treated with 78.1, 156.3, or 312.5 µM of Na_6_-**2** for 24 h, and then the cells were double-stained with FITC-Annexin V and PI and were exposed to a flow cytometry analysis. Dot plots show the distribution of early apoptotic (green A+ PI− square), late apoptotic (purple A+ PI+ square), and dead cells (red A− PI+ square). (**b**) A plot of fluorescent positive cells stained with dyes at different cluster concentration.

**Table 1 ijms-23-08734-t001:** M–P bond length analysis (M = Mo, W, and Re).

Cluster Compound	M–P Distances, (Average), Å	Refs.
H_6_-**1**·7H_2_O·2.5Et_2_O	2.522(2)–2.536(2)	this work
H_6_-**2**·3H_2_O·Et_2_O	2.524(2)–2.539(2)	
Na_6_-**1**·7.5H_2_O·Me_2_CO	2.538(3)–2.562(3)	
[{Mo_6_S_8_}(PPh_2_CH_2_CH_2_COOH)_6_]·4.5H_2_O·2.5Et_2_O	2.5444(9)–2.5594(8)	[44]
[{Mo_6_Se_8_}(PPh_2_CH_2_CH_2_COOH)_6_]·4.5H_2_O·2.5Et_2_O	2.544(2)–2.560(2)	
[{Re_6_Se_8_}(PPh_2_CH_2_CH_2_COOH)_6_]Br_2_·6H_2_O·Et_2_O	2.481(2)	[22]
Na_4_[{Re_6_S_8_}(PPh_2_CH_2_CH_2_COO)_6_]·4H_2_O	2.4842(8)–2.4913(7)	
Na_4_[{Re_6_Se_8_}(PPh_2_CH_2_CH_2_COO)_6_]·4H_2_O	2.4849(8)–2.4909(8)	

**Table 2 ijms-23-08734-t002:** The IC_5**0**_ values of [{M_6_Q_8_}(PPh_2_CH_2_CH_2_COO)_6_]^n^ (M = W, Re) clusters.

Cluster Complex	IC_50_ Value, μM	Refs.
Na_6_-**1**	1680 ± 70	this work
Na_6_-**2**	240 ± 20	
Na_4_[{Re_6_S_8_}(PPh_2_CH_2_CH_2_COO)_6_]	1150 ± 180	[22]
Na_4_[{Re_6_Se_8_}(PPh_2_CH_2_CH_2_COO)_6_]	730 ± 260	

## Data Availability

Crystal structure data can be obtained free of charge from The Cambridge Crystallographic Data Centre via www.ccdc.cam.ac.uk/data_request/cif accessed on 18 July 2022 or are available on request from the corresponding author.

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
