# Peer review of "Water-Soluble Chalcogenide W_6_-Clusters: On the Way to Biomedical Applications"

_ijms, 2022, doi:10.3390/ijms23158734_

Round 1

Reviewer 1 Report

Dear authors, 

After reading your manuscript I concluded that:

1. The clarity of the manuscript is not high. Proofreading is needed by a native speaker.

2. The introduction should contained more information about the scientific background. The references are limited. More work should be done here.

3. You didn't discussed your results. You need to discuss them in depth.

4. In conclusions you didn't mention any biological outcomes. Please kindly include them.

Best regards,

Author Response

Referee 1

Thank you for your positive feedback and relevant suggestions, which we address below.

Dear authors, 

After reading your manuscript I concluded that:

  1. The clarity of the manuscript is not high. Proofreading is needed by a native speaker.

Answer: The text of the article was improved.

  1. The introduction should contained more information about the scientific background. The references are limited. More work should be done here.

Answer: The introduction has been significantly extended in order to clearly show the scientific background on this topic.

  1. You didn't discussed your results. You need to discuss them in depth.

Answer: A more detailed discussion of the results has been added to the text of the article. Moreover, we performed an additional biological experiment to extend and clarify our results and discussion.

  1. In conclusions you didn't mention any biological outcomes. Please kindly include them.

Answer: The results of biological studies were included in the conclusions.

Reviewer 2 Report

The paper by Alena D. Gassanat all presents a reports of water-soluble chalcogenide W6-clusters - synthesis and physicochemical and biological properties. The issue covered in this paper is interesting, up-to-date and the paper is written clear and concise. The research methods used to evaluate properties of coumpounds characterized by X-ray structural analysis, NMR, IR, UV/vis spectroscopies, HR-MS spectrometry and electrochemical techniques are advantages of this article. Unfortunately, these methods do not provide information on how many percent the purity was. In my opinion the discussion of the results was properly described although in some places the discussion should be a bit more extensive.

I have comment and questions for the authors:

1.      Why chalcogenides W6-clusters are important? Please add in the introduction

2.      What is the yield of the synthesized compounds designated H6-1 and H6-2? Please add in the article

3.      What methods were used to purify both compounds? Please add in the article.

4.      The authors write that the compounds are well soluble in water. Are hydrogen interactions observed?

Therefore, I recommend publication of this manuscript after some minor revision that could additionally improve this paper.

Author Response

Referee 2

The paper by Alena D. Gassan at all presents a reports of water-soluble chalcogenide W6-clusters - synthesis and physicochemical and biological properties. The issue covered in this paper is interesting, up-to-date and the paper is written clear and concise. The research methods used to evaluate properties of coumpounds characterized by X-ray structural analysis, NMR, IR, UV/vis spectroscopies, HR-MS spectrometry and electrochemical techniques are advantages of this article. Unfortunately, these methods do not provide information on how many percent the purity was. In my opinion the discussion of the results was properly described although in some places the discussion should be a bit more extensive.

Answer: Thank you for your positive feedback and relevant suggestions, which we address below.

I have comment and questions for the authors:

  1. Why chalcogenides W6-clusters are important? Please add in the introduction

Answer: The required information was added in the introduction part.

  1. What is the yield of the synthesized compounds designated H6-1 and H6-2? Please add in the article

Answer: For compound H6-1 and H6-2 the yields are 69 and 75% correspondingly. The yields of compounds synthesized are mentioned in the 2.1 Synthesis and Characterization and 3.3. Synthetic procedures.

  1. What methods were used to purify both compounds? Please add in the article.

Answer: During isolation of compounds from the reaction mixture, the complexes were washed with DMF, acetonitrile, and water. These compounds do not require any additional purification methods (for example, chromatography). Purity of compounds was confirmed by a number of physicochemical methods of analysis which results are in good agreement with the proposed composition. Additional discussion of the process for isolating compounds from the reaction mixture has been added to the text.

  1. The authors write that the compounds are well soluble in water. Are hydrogen interactions observed?

Answer: Since the dissolution of the compounds requires deprotonation of the carboxyl groups of six ligands, it can be assumed that the compounds in solution will participate in hydrogen bonds with water molecules. An analysis of the crystal structure of Na6-1·7.5H2O·Me2CO showed that one carboxyl group (O12) participates in hydrogen bonds with disordered solvate water molecule O6W (O...O distances of 2.717 and 3.094 Å), located in the space between the chains formed by the sodium cations and cluster anions. Also, the cavities between the chains can be occupied by other solvent molecules (According to PLATON software, the void with the volume of 553 Å3 and about 111 electrons), which can be involved in hydrogen bonds with cluster anions. Additional discussion of crystalline structure has been added to the text.

Therefore, I recommend publication of this manuscript after some minor revision that could additionally improve this paper.

Round 2

Reviewer 1 Report

Dear authors,

your manuscript has been significantly improved and is ready for publication,

Best regards,